# Long-Term Effects and Potential Impact of Early Nutrition with Breast Milk or Infant Formula on Glucose Homeostasis Control in Healthy Children at 6 Years Old: A Follow-Up from the COGNIS Study

**DOI:** 10.3390/nu15040852

**Published:** 2023-02-07

**Authors:** Estefanía Diéguez, Ana Nieto-Ruiz, Natalia Sepúlveda-Valbuena, Florian Herrmann, Ahmad Agil, Roser De-Castellar, Jesús Jiménez, Hatim Azaryah, José Antonio García-Santos, Mercedes García-Bermúdez, Cristina Campoy

**Affiliations:** 1Department of Paediatrics, Faculty of Medicine, University of Granada, 18016 Granada, Spain; 2Instituto de Investigación Biosanitaria (ibs.GRANADA), Health Sciences Technological Park, 18012 Granada, Spain; 3EURISTIKOS Excellence Centre for Paediatric Research, Biomedical Research Centre, Faculty of Medicine, University of Granada, 18016 Granada, Spain; 4Nutrition and Biochemistry Department, Faculty of Sciences, Pontificia Universidad Javeriana, Bogotá 110231, Colombia; 5Department of Pharmacology, Faculty of Medicine, University of Granada, 18016 Granada, Spain; 6Federico Oloriz Neuroscience Institute, University of Granada, 18016 Granada, Spain; 7Ordesa Laboratories, 08830 Sant Boi de Llobregat, Spain; 8National Network of Research in Epidemiology and Public Health (CIBERESP), Institute of Health Carlos III (Granada’s node), 28029 Madrid, Spain

**Keywords:** early nutrition, growth velocity, continuous glucose monitoring, multiscale sample entropy, glucose coefficient of variation, glucose homeostasis, glycemic variability, body fat mass

## Abstract

There is scarce evidence about early nutrition programming of dynamic aspects of glucose homeostasis. We analyzed the long-term effects of early nutrition on glycemic variability in healthy children. A total of 92 children participating in the COGNIS study were considered for this analysis, who were fed with: a standard infant formula (SF, *n* = 32), an experimental formula (EF, *n* = 32), supplemented with milk fat globule membrane (MFGM) components, long-chain polyunsaturated fatty acids (LC-PUFAs), and synbiotics, or were breastfed (BF, *n* = 28). At 6 years old, BF children had lower mean glucose levels and higher multiscale sample entropy (MSE) compared to those fed with SF. No differences in MSE were found between EF and BF groups. Normal and slow weight gain velocity during the first 6 months of life were associated with higher MSE at 6 years, suggesting an early programming effect against later metabolic disorders, thus similarly to what we observed in breastfed children. Conclusion: According to our results, BF and normal/slow weight gain velocity during early life seem to protect against glucose homeostasis dysregulation at 6 years old. EF shows functional similarities to BF regarding children’s glucose variability. The detection of glucose dysregulation in healthy children would help to develop strategies to prevent the onset of metabolic disorders in adulthood.

## 1. Introduction

The period of lactation constitutes a window of opportunity, within the first 1000 days of life, to intervene and reduce lifelong metabolic disease risk. Glucose homeostasis depends on concerted functions of the brain, pancreas, hepatocytes, adipose tissue, etcetera. These organs are involved in maintaining glucose homeostasis and continue differentiation and growth during the lactation period. Thus, they are vulnerable to programming influences during the lactational window. As a matter of fact, overnutrition during this period could lead to hyperinsulinemia and impaired insulin signaling [1], and contributes to adipose hyperplasia [2].

Obesity and diabetes have become worldwide health problems during the last few decades, affecting more frequently the younger population [3,4]. It is well known that obesity is a causal risk factor for the development of an altered glucose metabolism and insulin resistance, preceding type 2 diabetes (T2D) [5,6,7], while weight loss has been associated with a lower incidence of T2D in individuals who already have impaired glucose tolerance [7]; thus, higher overall adiposity levels may promote insulin resistance. With the rising incidence of both diseases, identifying the windows of susceptibility that contribute to developmental programming are key to determine important time periods to target intervention strategies. In this sense, the lactational period has been identified as a critical window of opportunity for the establishment of body composition and insulin sensitivity because it is a period of greater plasticity of metabolic tissues [1].

Glycemic variability (GV) is the fluctuation of glucose levels above or under the normal range, which constitutes a key factor to take into account when evaluating the quality of glycemic control [8,9]. After at least 8 h of fasting, normal blood glucose levels are considered between 70–125 mg/dL [10]. GV increases progressively from pre-diabetes status to T2D [11]. Interestingly, higher glycemia and GV have been observed in older individuals compared with younger healthy ones [7], but studies on the potential association between adiposity and GV in non-obese and obese children without diabetes are still scarce. In the pediatric population, continuous glucose monitoring (CGM) devices constitute a suitable tool to obtain glucose data during the whole day in a less invasive way than blood glucose values and provide detailed information about glucose fluctuations, which could allow for the studying the association between body fat mass (BFM) and GV in this population [12].

Consequently, nutrition during the first 1000 days of life constitutes a window of opportunity in the development of future health outcomes since it is a key factor in the prevention and protection against different metabolic disorders later in life [13]. In this line, an intrauterine environment with an excess of glucose might promote adipogenesis, resulting in a long-term metabolic risk to the offspring [13]. Thus, changes both in quantity and quality of nutrients during this period may permanently influence organs’ maturation and function, better known as ‘early nutrition programming’ [14]. There is wide evidence that early postnatal nutrition can program a lifelong obesity risk, independently of in utero exposure. In fact, several epidemiological studies have linked early infant nutrition (infant formula vs. breastfeeding) to adult obesity risk [15]. Indeed, longer breastfeeding duration has been associated with a reduced risk of obesity later in life [15]. Breast milk is considered the gold standard of infant nutrition because of its composition in bioactive nutrients, and associated short- and long-term health benefits [16,17]. Nevertheless, breastfeeding sometimes is not possible. Thus, it is necessary to supplement infant formulas with different bioactive nutrients in order to close the nutritional gap with human breast milk. Among these nutrients, dietary lipids constitute the main source of energy in infants, providing 45–55% of the total energy during the first 6 months of life. Thus, the quality and quantity of dietary lipids in infant formulas are very important if we consider their impact on health outcomes later in life [18]. As a matter of fact, fish and omega-3 polyunsaturated fatty acids (n-3 PUFAs) supplements could help reduce the risk of metabolic diseases [19]. Nevertheless, there is still scarce evidence about the long-term effects of early nutrition programming of dynamic aspects of glucose homeostasis related to obesity and increased adiposity, which could constitute a target for disease risk reduction. 

In the present study we aimed to analyze the long-term effects of early nutrition on glycemic variability (CGM data) and BFM in healthy children, participating in the COGNIS study. These children were randomly assigned to receive, during their first 18 months of life, an experimental infant formula (EF) or a standard formula (SF); furthermore, children who were breastfed (BF), were included in the analysis as control group. As a secondary objective, we analyzed the relationship between CGM and BFM data with dietary intake at 6 years old.

## 2. Materials and Methods

### 2.1. Ethics, Informed Consent, and Permissions

The COGNIS study followed the updated Declaration of Helsinki II Principles [20,21]. The Research Bioethical Committee from the University of Granada (Granada, Spain), and the Bioethical Committees for Clinical Research of San Cecilio University Clinical and University Mother-Infant Hospitals of Granada (Granada, Spain) approved the project and protocols. All families were informed about protocols and a signed written consent was obtained from each parent or legal guardian before involving each child in the study. 

### 2.2. Study Design and Subjects

The COGNIS study (Clinical Trial Registration: www.ClinicalTrials.gov, identifier: NCT02094547) is a prospective, randomized, and double-blind clinical trial with a nutritional intervention using an infant formula supplemented with different bioactive nutrients. Detailed information on this study was described elsewhere [22,23,24]. Briefly, a total of 220 healthy Spanish infants were included; of them, 170 were randomized to receive, during their first 18 months of life, either a standard infant formula (SF: *n* = 85) or an experimental infant formula (EF: *n* = 85) enriched with bioactive components, such as milk fat globule membrane (MFGM) components [10% of total protein (wt:wt)], long-chain (LC)-PUFAs [arachidonic acid (ARA) and docosahexaenoic acid (DHA)], gangliosides, nucleotides, synbiotics [mix of fructooligosaccharides and inulin (ratio 1:1), *Bifidobacterium infantis* IM1 and *Lactobacillus rhamnosus* LCS-742], and sialic acid. Additionally, 50 healthy breastfed (BF) infants were included as a control group. 

After drop-outs, 110 children attended the follow-up call at 6 years of age (SF: *n* = 39; EF: *n* = 39; BF: *n* = 32); of them, 92 had valid CGM data (SF: *n* = 32; EF: *n* = 32; BF: *n* = 28). A detailed flowchart from the baseline visit to 6 years old is shown in Figure 1.

### 2.3. Demographical and Clinical Baseline Characteristics

Parents’ baseline characteristics, such as maternal and paternal age, socioeconomic status, educational level, place of residence, and intelligence quotient (IQ), were collected at study entry. Information about pre-gestational body mass index (pBMI), gestational weight gain (GWG), siblings, type of delivery, and smoking during pregnancy were also registered. Neonatal information, including gestational age, sex, and anthropometric characteristics at birth [weight, length, and head circumference (HC)], were obtained from clinical records.

### 2.4. Anthropometric Measures 

Anthropometric data, including weight, height, BMI, HC, as well as tricipital and subscapular skinfolds, were obtained by a trained nutritionist at the children’s 6 years old follow-up visit, following the World Health Organization (WHO) and the International Society for the Advancement of Kinanthropometry (ISAK) standard procedures [26,27]. Weight was measured using Tanita Body Composition Analyzer BC-418MA^®^ (Biologica TM S.L., Barcelona, Spain); height was obtained using SECA stadiometer264, max 220 cm; skinfolds measures were obtained using Holtain Model DIM-98610ND, max 40 mm. HC was measured using SECA 212 measuring tape, max 59 cm. Weight, height, and BMI *z*-scores were calculated according to the WHO growth standard charts by age and sex, using the WHO Anthro Plus software package version 3.2.2 (World Health Organization, Geneva, Switzerland) [26]. 

#### 2.4.1. Growth Velocity and Catch-Up

Growth velocity was calculated according to weight and length gains per day. These were calculated at three different time intervals: (1) from birth to 6 months of life, (2) from 6 to 12 months of life, and (3) from 12 to 18 months of life. These data were compared using the WHO growth standards and classified as described in previous studies [24], using the WHO Anthro software package version 3.2.2 (World Health Organization, Geneva, Switzerland). Catch-up growth was also calculated as weight for age *z*-score (WAZ) and weight for length *z*-score. Differences in *z*-scores were calculated at the three different time intervals mentioned above and classified as mentioned in previous research [24]. 

#### 2.4.2. Body Fat Mass (BFM)

To analyze BFM distribution, skinfolds measures and bioelectrical impedance with TANITA^®^ (Biologica TM S.L., Barcelona, Spain) were performed: a.BFM percentage calculated with the triceps and subscapular skinfolds using the Slaughter’s equations [28], as follows:

When the skinfold summation was less than 35 mm, the equation used for boys was 1.21 *×* (triceps + subscapular) *−* 0.008 × (triceps + subscapular)^2^
*−* 1.7, and 1.33 × (triceps + subscapular) *−* 0.013 × (triceps + subscapular)^2^
*−* 2.5 for girls. 

When the skinfold summation was higher than 35 mm, the equation used for boys was 0.783 *×* (triceps + subscapular) + 1.6, and 0.546 *×* (triceps + subscapular) + 9.7 for girls. 

Once the BFM percentage was calculated, children were classified by sex and age using BFM percentile values in European children [29] in either of the following groups, thinness (≤P3), normoweight (NW) (>P3 and <P90), or excess weight (EW) (≥P90 overweight and ≥P97 obese), which included both overweight and obese children. 

b.BFM percentage calculated by Slaughter’s equations was corroborated by Tanita Body Composition Analyzer BC-418MA^®^ (Biologica TM S.L., Barcelona, Spain), which indirectly measures total body water, fat mass, and fat-free mass using a high-frequency current (50 kHz, 90 µA) via 8-electrode. This method is based on the principle that body water conductivity changes in different body compartments [30,31]. Once BFM was obtained, children were classified by sex and age using percentile values according to the McCarthy’s tables (2006) [32], in the following groups: thinness (≤P2), NW (>P2 and <P85), or EW (≥P85 overweight and ≥P95 obese). The latter group included both overweight and obese children.

### 2.5. Continuous Glucose Monitoring (CGM) 

At 6 years old, children’s glucose homeostasis was evaluated by a 24 h CGM device for an average of 7 days. Glucose levels were measured with the FreeStyle Glucose FlashMonitoring System (http://www.freestylelibre.es; Reference 0086, Abbott Laboratories S.A., Granada, Spain). This system sensor consists of a small-sized device that fits on the arm and measures glucose levels in the interstitial fluid of subjects of at least 4 years of age at any time by using a reader that scans measurements instantaneously [33]. Parents were instructed on how to use it by trained personnel, and they were told to scan the sensor at least before and right after every meal and two hours after eating. The FreeStyle LibreLink software (version 2.4.1, Abbott Laboratories S.A.) was used to download glucose data, including mean glucose data, number of low glucose events, as well as graphics of daily glucose pattern [33]. 

Glycemic variability (GV) for each child was assessed using the glucose coefficient of variation (CV) and the multiscale sample entropy (MSE) approach, on data obtained from the CGM device. Glucose CV was calculated using the equation (SD of mean glucose levels/mean glucose levels), and to express these data in percentage, we multiplied this equation by 100. It is a very useful parameter to measure GV in the diabetic population [34]. With the MSE approach, we obtained measures of sample entropy at various time series with R software (CGManalyzer package version 1.3). In order to adjust the data, due to MSE not being a non-linear variable, it is displayed in a time series of 3 min from 3 to 30 min; thus, equal space between any two consecutive time points can be achieved (equalInterval.fn) [35,36,37]. CGManalyzer also has a function to fix missing values when necessary (fixMissing.fn) [35]. Sample entropy measures the irregularity and complexity of physiological signals. Lower values of sample entropy imply higher regularity in a time series, while higher values imply substantial fluctuation [36,37]. 

### 2.6. Dietary Intake 

To collect information about participants’ dietary intakes at 6 years old, three-day dietary records were used based on the Food and Agriculture Organization of the United Nations (FAO) methods [38]. Dietary intake information was collected concomitantly with CGM data. DIAL software (Alce Ingeniería, Madrid, Spain) [39] was used to convert food consumption data into macro- and micronutrient intakes. Nutrient intake was analyzed according to the dietary reference intakes (DRIs) [40], in order to evaluate whether the dietary intake was deficient, adequate, or excessive according to the recommendation, taking into account age and sex (see Appendix A). Acceptable macronutrient distribution ranges (AMDR) [40] were also calculated and classified as deficient, adequate, or excessive according to the recommendation, by age (see Appendix A). AMDR represents the percentage of energy that each macronutrient (carbohydrates, proteins, or lipids) supplies to the total daily energy intake.

### 2.7. Statistical Analysis

Statistical analysis of the participants’ baseline characteristics was performed using IBM^®^ SPSS Statistics^®^ program, version 25.0 (SPSS Inc. Chicago, IL, USA). Normally distributed variables were presented as mean and standard deviation (SD), and non-normal variables as the median and interquartile range (IQR). Categorical variables were shown as frequencies and percentages. The following tests were performed: analysis of variance (ANOVA) or Welch for normally distributed variables, Kruskal–Wallis test for non-normal continuous variables, and Chi-square or Fisher test for categorical variables. In the event of significant group differences, Bonferroni corrected post hoc comparisons were used to identify significant pair-wise group differences (corrected *p*-values < 0.05). These analyses were adjusted by the following confounding variables: maternal age, parents’ educational level, and socioeconomic status. 

Partial correlations were carried out to study the association between continuous glucose monitoring data, dietary intake, and anthropometric data at 6 years old. Correlations were adjusted by the same variables, maternal age, parents’ educational level, and socioeconomic status.

To analyze continuous glucose monitoring data, R software (version 4.1.2, package CGManalyzer) was used. An MSE approach was carried out obtaining measures of sample entropy at various temporal scales. *p*-values < 0.05 were considered statistically significant.

## 3. Results

### 3.1. Baseline Characteristics of the Six-Year-Old Children Participating in the COGNIS Study and Their Parents

The baseline characteristics of children and their parents participating in the COGNIS study are shown in Table 1. There were statistically significant differences between study groups regarding maternal age, parents’ educational level, and socioeconomic status. Mothers of BF infants were significantly older (*p* = 0.015) than mothers of EF infants and had higher educational level (*p* = 0.002) compared to both infant formula groups (SF and EF). Fathers of BF infants had higher educational level (*p* = 0.005) compared to the EF group. Parents of BF infants had higher socioeconomic status (*p* = 0.004) compared to both formula groups. However, parents of the three study groups showed similar IQ. Similarly, mothers had similar pBMI and GWG, and they were usually non-smokers and did not develop gestational diabetes mellitus. Infants were born more frequently by vaginal delivery. 

Concerning neonatal anthropometric characteristics (weight, length, and HC), there were no differences between the study groups.

Regarding children’s anthropometric characteristics at 6 years old, no differences were found between study groups (Table 2). Thinness was not considered according to skinfolds’ BFM classification in the following analyses because there was only one child in the SF group that was classified as thin. On the other hand, there were no children measured by TANITA^®^ classified as thin. Finally, overweight and obesity were defined as EW in the following analyses, as mentioned above, in the Materials and Methods section.

### 3.2. Multiscale Sample Entropy (MSE) Analysis in Six-Year-Old Children

#### 3.2.1. MSE Analysis Considering Study Groups at 6 Years Old

The baseline glucose data at 6 years old, considering the study groups, are shown in Table 3. BF group children had significantly lower mean glucose levels and adjusted mean glucose levels compared to SF group children (*p_adj_* = 0.026; *p_adj_* = 0.005, respectively). Nonetheless, when we considered the minimum and maximum means of glucose levels, we observed similar minimum levels between the three study groups, and higher maximum levels in the EF group compared to the BF group (*p* = 0.045) (Figure 2). Adjusted mean glucose levels were the glucose data displayed in a time series, specifically in 3 min time series from 3 to 30 min, so that equal space between any two consecutive time points can be achieved. The multiscale sample entropy (MSE) increment at 3–30 min was statistically significant between study groups, but after adjusting for the confounding variables, maternal age, parents’ educational level, and socioeconomic status, significance was lost. Finally, glucose coefficient of variation (CV) was lower in BF children compared to EF ones (*p_adj_* = 0.014). 

Afterwards, MSE expressed as mean and SD was calculated in each study group, as well as *p*-values after comparing groups in pairs (SF vs. EF, SF vs. BF, EF vs. BF) for different glucose time series, as shown in Table 4. There were no statistically significant differences between formula groups, nor EF compared with BF children. However, SF and BF presented statistically significant differences in MSE at 9, 12, 15, 18, 21, 24, and 30 min of the time series, being higher in the BF group compared to the SF group (*p* = 0.045; *p* = 0.034; *p* = 0.048; *p* = 0.037; *p* = 0.016; *p* = 0.045; *p* = 0.021, respectively) (Table 4). 

#### 3.2.2. MSE Analysis in Children according to Catch-Up and Growth Velocity during the First Months of Life

After studying MSE according to catch-up growth, we did not find any significant results, nor with weight for age *z*-score (WAZ) or weight for length *z*-score. Additionally, MSE was analyzed considering growth velocity calculated according to length and weight gains. Nonetheless, we did not find any statistically significant data from birth to 18 months of life regarding growth velocity according to length gain. 

Finally, we did not find significant data from 6 to 18 months of age regarding growth velocity according to weight gain, but we did find significance from birth to 6 months of life. Regarding adjusted mean glucose levels, we found no differences between the three groups (normal, rapid, and slow). Nonetheless, we found higher MSE in normal compared to rapid weight gain velocity children at 3, 9, 12, 15, 18, 21, 24, and 27 min (*p* = 0.026; *p* = 0.045; *p* = 0.037; *p* = 0.025; *p* = 0.030; *p* = 0.019; *p* = 0.022; *p* = 0.045, respectively). Furthermore, we found higher MSE in slow compared with rapid weight gain velocity children at 3, 6, 9, 12, 15, 18, 21, and 24 min (*p* = 0.023; *p* = 0.047; *p* = 0.045; *p* = 0.026; *p* = 0.029; *p* = 0.025; *p* = 0.017; *p* = 0.017, respectively). Similar MSE values were found between normal and slow weight gain velocity children (Table 5). 

#### 3.2.3. MSE Analysis in Six-Year-Old Children Considering Study Groups according to Their BFM Calculated Using the Slaughter’s Equations

Afterwards, to know whether the type of feeding received during the first 18 months of life could affect glucose homeostasis in children aged 6 years, an MSE analysis was performed for each study group according to the BFM percentage calculated with the Slaughter’s equations (Table 6 and Appendix A).

We compared normoweight (NW) children by study group (Table 6), as well as excess weight (EW) children (Appendix A). NW children who received SF had significantly lower MSE compared to NW breastfed children from 12 to 30 min (*p* = 0.012; *p* = 0.011; *p* = 0.006; *p* = 0.003; *p* = 0.007; *p* = 0.006; *p* = 0.002, respectively) (Table 6). Nonetheless, we did not find any differences between study groups in EW children (Appendix A). 

#### 3.2.4. MSE Analysis in Six-Year-Old Children without Considering Study Groups according to Their BFM Calculated Using the Slaughter’s Equations

After classifying the study population by BFM percentage calculated using the Slaughter’s equations (NW: *n* = 68 and EW: *n* = 22), we observed that adjusted mean glucose levels were significantly higher in NW children compared to EW (NW: 100.15 ± 9.16; EW: 94.91 ± 9.45; *p* = 0.029). Nonetheless, these glucose values were within the normal range in both groups. Moreover, significant differences in MSE were also found between both BFM groups. In fact, higher MSE in the EW group compared to the NW group only at three and six time series (*p* = 0.047; *p* = 0.045, respectively). However, significance disappeared from nine to thirty time series.

#### 3.2.5. MSE Analysis in Six-Year-Old Children Considering Study Groups according to Their BFM Measured by Bioelectrical Impedance (TANITA^®^)

To corroborate data obtained with skinfolds, we next classified the study population by BFM percentage measured with TANITA^®^ (Biologica TM S.L., Barcelona, Spain) and according to their type of feeding during the first 18 months of life. Regarding adjusted mean glucose levels, we did not find any significant differences in the SF group (NW: 102.90 ± 9.94; EW: 99.71 ± 8.68; *p* = 0.36), the EF group (NW: 99.99 ± 10.33; EW: 99.80 ± 6.66; *p* = 0.95), or the BF group (NW: 97.62 ± 5.54; EW: 90.54 ± 12.59; *p* = 0.14). It is worth noting that these adjusted mean glucose values were within the normal range. Similarly, there were no statistically significant differences between the three groups regarding MSE and BFM measured with bioimpedance.

#### 3.2.6. Children’s MSE Analysis according to Their BFM Measured with TANITA^®^

When we classified the study population by BFM percentage measured with TANITA^®^ (NW: *n* = 58; EW: *n* = 30), we found no statistically significant differences in adjusted mean glucose levels between groups (W: 100.39 ± 9.20; EW: 96.99 ± 10.09; *p* = 0.13). Once again, adjusted mean glucose values were within the normal range in both groups. Accordingly, no statistically significant differences were found in MSE between NW and EW children. 

### 3.3. Dietary Intake Analysis in COGNIS Children at 6 Years Old

Next, analysis of dietary intake in COGNIS children aged 6 years was performed (Appendix A). We found that the energy supplied by simple sugars to total daily energy intake (AMDR) was significantly higher in the BF group compared to the EF group (*p* = 0.040) (Figure 3). This significance was maintained after adjusting by confounder factors, including maternal age, parents’ educational level, and socioeconomic status (*p_adj_* = 0.017) (Figure 3). Nonetheless, after comparing simple sugars AMDR to DRI, no statistical difference between study groups was found (*p* = 0.085) and adequate simple sugars AMDR according to DRI was observed (Appendix A). Additionally, there were no statistically significant differences between study groups in simple sugars intake (*p* = 0.17), even after adjusting by confounding variables (*p_adj_* = 0.19) (Appendix A). Regarding eicosapentaenoic acid (EPA) and DHA intakes (g/day), the EF group had higher intakes compared to the SF group (*p* = 0.006; *p* = 0.008, respectively); after adjusting by confounding variables, there were no differences (*p_adj_* = 0.057; *p_adj_* = 0.053, respectively) (Appendix A). 

### 3.4. Association between Anthropometric Measures, Glucose Data, and Dietary Intake at 6 Years Old

Partial correlation analyses were carried out to evaluate potential associations between anthropometric, glucose data, and dietary intake in children at 6 years old in the whole study population. 

As shown in Figure 4, BMI for age *z*-score (BAZ) had a positive correlation with protein AMDR (%) (*r* = 0.293, *p_adj_* = 0.005). Height for age *z*-score (HAZ) also had a positive correlation with protein AMDR (%) (*r* = 0.213, *p_adj_* = 0.042), docosapentaenoic acid (DPA) (*r* = 0.245, *p_adj_* = 0.019), and docosaexaenoic acid (DHA) intakes (g/day) (*r* = 0.222, *p_adj_* = 0.033). Skinfolds’ BFM (%) also showed a positive correlation with protein AMDR (%) (*r* = 0.246, *p_adj_* = 0.018), as well as TANITA^®^ BFM (%) with protein AMDR (%) (*r* = 0.338, *p_adj_* = 0.001) (Figure 5). Finally, glucose coefficient of variation (CV) showed a positive correlation with total carbohydrates intake (g/day) (*r* = 0.302, *p_adj_* = 0.006) (Figure 6).

Conversely, a negative correlation was found between adjusted mean glucose levels (mg/dL) and total protein (*r* = −0.222, *p_adj_* = 0.047), total lipids (*r* = −0.229, *p_adj_* = 0.039), and saturated fatty acids (SFAs) (*r* = −0.284, *p_adj_* = 0.010) intakes (g/day) (Figure 6). 

## 4. Discussion

Studies on the association between continuous glucose monitoring (CGM) data, body fat mass (BFM), and dietary intake in healthy children are still limited and as far as we know, no long-term clinical trials have been carried out. Our results showed lower mean glucose levels and adjusted mean glucose levels in BF children compared to SF ones. Regarding glucose coefficient of variation (CV), it was lower in BF children compared to EF children. However, despite these lower glucose levels, we observed higher MSE in BF children compared to SF ones. Nonetheless, we should take into account that glucose levels were within the normal range in the three groups, and glucose CV was below 20% in all groups. Glucose CV below 36% is associated with low glycemic variability in diabetic patients [34]. Indeed, glucose CV is a useful tool in the management of the diabetic population since it allows to differentiate between patients with high or low glycemic variability (GV) [34]. 

Considering that MSE measures the irregularity and complexity of physiological signals, such as glucose levels, it might be useful for the diagnosis and prognosis of different diseases, such as diabetes. Lower values of sample entropy imply higher regularity in a time series, while higher values imply substantial fluctuation in diabetic patients [36,37]. However, it is not clear the real significance of MSE in healthy children. In this regard, there were no statistically significant differences between formula groups, or EF compared with BF children, but MSE resulted higher in BF children compared to those from the SF group. These results suggest greater similarity regarding glucose homeostasis between children fed with EF and those who were BF. Once again, it is important to highlight that we are studying healthy children and glucose CV was way below 36%, indicating low GV, as expected in a healthy population. Moreover, it has been shown that BF can reduce the risk of diabetes in childhood. Curiously, previous research has shown that exclusively BF infants have lower insulin, and formula-fed infants have higher postprandial plasma insulin levels and a prolonged insulin response compared to BF infants [41], which could lead to the development of insulin resistance and later on, to the onset of diabetes. This could explain why BF children had higher MSE compared to SF children. 

Something similar happens with other metabolic regulations during early life, such as cholesterol and breast milk. In fact, mechanisms underlying the association between breastfeeding and lower cholesterol levels in adulthood induces nutritional programming wherein early exposure to exogenous cholesterol (higher levels in human milk), which suppresses endogenous synthesis of cholesterol through downregulation of hepatic hydroxymethyl glutaryl coenzyme A (HMG CoA) reductase [42]. On the other side, the carbohydrates in most formulas are simple sugars, such as corn syrup solids, which can alter endogenous cholesterol programming. One suggested potential mechanism underlying this relationship is glucose- and insulin-mediated increases in PCSK9 (proprotein convertase subtilisin/kexin type 9) of LDLR protein, setting the stage for a cycle of increasing LDL [43]. Then, it could be possible that functional similarities found here between EF and BF children in terms of glucose homeostasis regulation may be associated to an early programming effect of breast milk, which is partially mimicked by the EF supplemented with some functional components present in human milk (MFGM components, LC-PUFAs, synbiotics, sialic acid, nucleotides, etcetera). Nevertheless, these results should be taken with caution, and more studies are needed to demonstrate this hypothesis.

According to previous results from the COGNIS study [24], we continue exploring the relationship between early nutrition and growth velocity and catch-up growth during the first months of life, but in this case in association with glucose homeostasis. From birth to 6 months of age, we found higher MSE in those children who had normal weight gain velocity (NWGV) compared to those showing rapid weight gain velocity (RWGV). Furthermore, we found higher MSE in children with slow weight gain velocity (SWGV) compared to children who showed RWGV in the first 6 months of life, while similar MSE data were found between NWGV and SWGV children. Infant weight gain is known as the primary indicator of healthy growth; higher weight gains between 3 and 12 months of age have been related to higher risk of obesity and other metabolic disorders [24]. NWGV and SWGV children presented higher MSE, which suggests again an early programming effect of slower growth velocity against later metabolic disorders, thus similarly to what we observed in BF children. 

Afterwards, we classified the study population according to BFM considering the three study groups. Regarding skinfolds’ BFM, we compared NW children by the COGNIS study group. Higher MSE was observed in BF children compared to SF children, but we did not find any differences when comparing COGNIS study groups in EW children. BFM percentage using bioelectrical impedance (TANITA^®^, Biologica TM S.L., Barcelona, Spain) was also assessed, but no significant differences were found between the three study groups regarding MSE and BFM. 

Studying the whole population, higher MSE was found at 3 and 6 min in the EW group compared to the NW group (classified according to the Slaughter equations), but the significance disappeared from 9 to 30 min. Furthermore, no significant differences were found in MSE between NW and EW children when classifying by BFM% obtained with TANITA^®^ (Biologica TM S.L., Barcelona, Spain). Thus, it is necessary to carry out more longitudinal studies with higher sample size to corroborate these results, since it is well known that obesity is a risk factor for the development of metabolic disorders, such as type 2 diabetes mellitus [44].

Most of the studies carried out with CGM devices have been performed in the diabetic population with the goal to achieve an adequate glycemic control; however, only few studies have been carried out in the healthy pediatric population. Nonetheless, CGM devices could be very useful to study early glucose patterns in healthy children to prevent future metabolic diseases and their associated health burdens. In a study carried out in 26 non-diabetic healthy weight or overweight/obese children aged 7 to 12 years, authors compared the FreeStyle Libre Pro CGM device with plasma glucose during a 2 h oral glucose tolerance test. Children participating in the study were classified according to the BMI by sex and age; in contrast with our study where we classified the children as thin, normal, or excess weight (overweight or obese) according to skinfolds and bioelectrical impedance (more reliable measures than BMI, since skinfolds and bioelectrical impedance reflect body fat distribution). Participants wore the device for 6 days, similar to our study, finding that among those children without diabetes, the CGM device was well tolerated, and the results were consistent with plasma glucose levels after the oral glucose tolerance test. Nonetheless, there were significant differences regarding fasting glucose, and the CGM device seemed to underestimate plasma glucose in those subjects with overweight/obesity [45]. Thus, in our study, EW children high glucose values could have been underestimated. Nonetheless, once again it is necessary to carry out more studies in non-diabetic pediatric populations to corroborate these results.

Increased adiposity is known to be a risk factor for suboptimal diabetes control. In a study with overweight and obese children with type 1 diabetes (T1D), aged less than 21 years and optimal glucose control, it was confirmed that being overweight was associated with suboptimal glucose control; even though, the use of CGM devices and frequent blood glucose checks between overweight and obese participants compared to the lean ones was the same [46]. Nonetheless, in our study we did not find any conclusive results regarding MSE and BFM percentage so the results should be taken with caution; furthermore, T1D children were classified according to BMI by age and sex [46], which is a less reliable measure compared to skinfolds and bioimpedance, used to measure BFM in our study.

Finally, in an eighteen-month randomized controlled trial with 136 participants with T1D, aged 8 to 17 years, body composition was measured by dual energy X-ray absorptiometry and BMI, while GV was measured during 3 days by a CGM device together with glycosylated hemoglobin (HbA1c). In contrast with the COGNIS study, where participants are non-diabetic, a short-term GV was detected, rather than long-term obtained with HbA1c. These short-term fluctuations were measured in our study using the CGM device, but during an average of 7 days rather than only 3 days. Authors found that greater BMI and adiposity were related with increased hyperglycemic events [47], while in our study we found non-conclusive results, perhaps because we studied healthy children. Thus, again, more studies are necessary to corroborate these results. 

Taking into account all mentioned above, one of the most important interventions for improving glucose homeostasis is the diet, especially an early diet [48]. There is scientific evidence that excessive protein consumption during early life leads to an increased insulin concentration, which promotes adipose tissue deposition and the risk of overweight, obesity, and type 2 diabetes in the subsequent years [41]. Thus, early interventions may be crucial to prevent diabetes development, since it has been observed that overweight or obesity solely are enough to cause insulin resistance and GV [49], which could lead to the development of diabetes later in life. In addition, it has been demonstrated that the carbohydrate content of a meal and the glycemic index (GI) of the carbohydrate consumed determine the postprandial glycemic response. High-GI carbohydrates diets have been shown to be risk factors for diabetes onset, while low-GI carbohydrates diets contribute to weight loss and improving insulin action and glucose tolerance in obese insulin-resistant individuals [50]. Having in mind these considerations, we studied dietary intake in COGNIS children at 6 years old. We found that simple sugars AMDR was significantly higher in the BF group compared to the EF group. Nevertheless, after comparing simple sugars AMDR to DRI, there was no difference between study groups. Additionally, there were no differences between study groups regarding simple sugars intake. On the other hand, at 6 years of age, children from the EF group had higher EPA and DHA intakes (g/day) compared to those from the SF group, though after adjusting by confounding variables, this resulted to be not statistically significant.

Interestingly, after comparing BFM and glucose data with dietary intake in the whole sample population, we observed a positive correlation between BMI for age *z*-score (BAZ), height for age *z*-score (HAZ), skinfolds’, and bioelectrical impedance BFM (%) with protein AMDR (%). These results agree with the concept already mentioned, that high protein consumption promotes adipose tissue deposition and a higher risk of developing overweight and obesity [41]. Nonetheless, according to our results, the three study groups had an adequate protein AMDR when we compared it with the DRIs. Additionally, we observed a positive correlation between HAZ, and DPA and DHA intakes (g/day). Plenty epidemiological studies have shown benefits of n-3 PUFAs on child health, positively affecting children’s growth [51]. Finally, glucose CV showed a positive correlation with total carbohydrates intake (g/day); thus, higher carbohydrates intake, especially simple sugars, would lead to higher glycemic variability. 

Conversely, a negative correlation was found between adjusted mean glucose levels (mg/dL) and total protein, total lipids, and saturated fatty acids (SFAs) intakes (g/day). Therefore, higher total protein, total lipids, and SFAs intakes might be associated with lower mean glucose levels. Amino acids have been shown to modulate glucose homeostasis by modulating insulin release [52], while animal studies have shown beneficial effects of lipids, in particular n-3 PUFAs, on insulin sensitivity and weight loss. Due to these effects on weight loss in obese rodents, it is difficult to know if n-3 PUFAs have direct effects on insulin sensitivity. Nonetheless, EPA intake has been shown to improve insulin sensitivity in obese mice, despite similar body weights [53]. Regarding SFAs, according to recent studies, they could have a limited role in the development of metabolic syndrome because as long as SFAs intake is associated with healthy eating patterns, it is not necessarily associated with negative health outcomes [54]. It is important to highlight that beneficial effects of protein and lipids intake on BFM and mean glucose levels are a product of the synergistic effect of all nutrients. Thus, a better knowledge about the impact of individual foods and nutrients on health is still challenging because of the numerous food–nutrient interactions. 

The main strength of the present study is its design as a randomized, double-blind long-term longitudinal study. In contrast with most of the studies mentioned above, we used skinfolds and bioimpedance as measures of adiposity rather than BMI alone, which is correlated with body fat, but as an indirect measure. In fact, BMI does not reflect body fat distribution like the skinfolds or TANITA^®^ (Biologica TM S.L., Barcelona, Spain), which constitute more appropriate and reliable measures of BFM%. Nonetheless, we included in our analyses BMI and BAZ as well to have a more complete picture. Another strength is that children wore the CGM device an average of 7 days, which allowed us to collect continuous glucose data throughout the day and during the night, rather than only a single fasting glucose measure. It is worth noting that these glucose measurements are from the interstitial fluid and not blood glucose levels; in this regard, studies have demonstrated the reliability of CGM devices compared to blood glucose levels in children with diabetes [46,55] and without diabetes [45].

Nevertheless, this study has limitations that are worth mentioning; the first one being that the small sample size was due to the drop-out during the 6 years of follow-up, and the lack of availability of more data, due to not all parents that came to the 6 years old follow-up visit with their children wanting them to wear the 24 h CGM device. However, most of the few studies carried out in non-diabetic young children have a smaller sample size compared to our study [56,57]. 

In conclusion, there is scarce evidence about early nutrition programming of dynamics aspects of glucose homeostasis, which together with lifestyle interventions could reduce risks of developing non-communicable diseases. Our findings suggest that the type of feeding and growth velocity during early life might be associated with glucose homeostasis control at 6 years old. As a matter of fact, BF seems to have a programming effect protecting against the development of glucose homeostasis dysregulation. At 6 years of age, there were no differences in MSE between EF and BF children, suggesting functional similarities between them. However, despite the higher MSE, BF children had lower mean glucose levels compared to children from the SF group. 

On the other side, growth velocity during the first 6 months of life seems to have a role in later glucose homeostasis during childhood. At 6 years of age, children who showed normal and slow weight gain velocity during their first 6 months of life presented higher MSE, suggesting an early programming effect of slower growth velocity against later metabolic disorders, thus similarly to what we observed in BF children. Nonetheless, in the present study, we did not find any conclusive data regarding long-term effects of early nutrition on adiposity, but as expected, daily intake of proteins, carbohydrates, and lipids at 6 years of age showed significant associations with glucose levels and glucose CV. 

The present results suggest that the improvement of infant formulas with bioactive compounds puts them closer to human milk functionality. Furthermore, it is highlighted that detection of glucose dysregulation in healthy children would help to develop early strategies, such as prompt dietetic interventions, to prevent metabolic disorders (i.e., type 2 diabetes) later in life. Longitudinal studies in healthy children including early nutritional intervention are lacking; thus, new studies with greater sample sizes are needed to corroborate our findings. 

## Figures and Tables

**Figure 1 nutrients-15-00852-f001:**
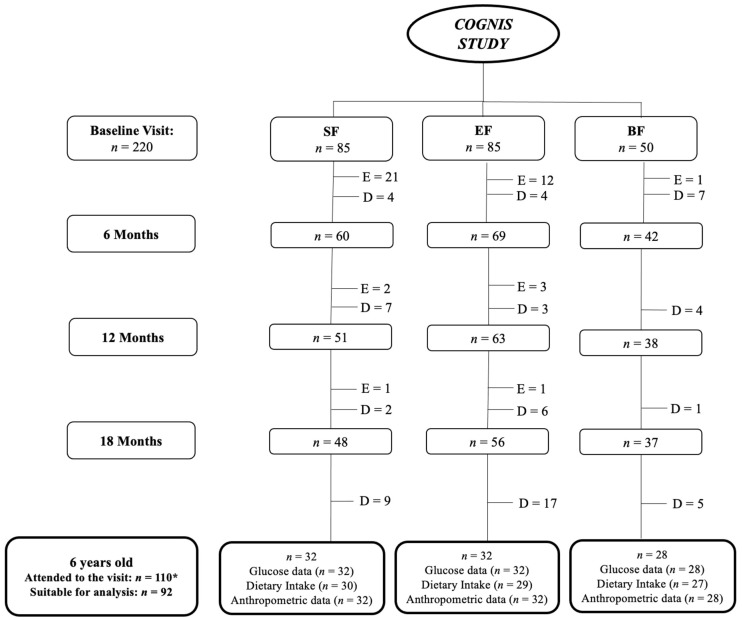
Participant flowchart from baseline visit to 6 years old. BF: breastfeeding; D: drop-outs; E: exclusions; EF: experimental infant formula; *n*: sample size; SF: standard infant formula. Up to 18 months of life, a total of 40 infants were excluded in the SF and EF groups as previously described [25]: 24 were excluded in the SF group (1 infant due to perinatal hypoxia, 1 infant had growth restriction, not related to the infant formula, 15 infants did not take the infant formula, 2 had infant colic, 3 were excluded due to lactose intolerance, 1 infant due to digestive surgical intervention, and 1 infant suffered hydrocephalia); 16 infants were excluded in the EF group (2 infants presented growth restriction, not related to the infant formula, 2 infants had lactose intolerance, 11 infants did not take the infant formula, and 1 was excluded due to epileptic seizure). While in the BF group, one infant was excluded, because he was not exclusively breastfed beyond 2 months of age. In the follow-up visits, drop-outs were due to the participants that decided not to continue in the study; then, 110* children (SF: 39; EF: 39; BF: 32) attended the follow-up visit at 6 years old. Nonetheless, not all parents attending the follow-up visit wanted their children to wear the 24 h continuous glucose monitoring (CGM) device. Mean glucose data were collected with a CGM device for an average of 7 days. Those glucose data registered for less than three days were not included in the final analysis. Lastly, at 6 years old, 92 children were included in the current analysis (SF: 32; EF: 32; BF: 28).

**Figure 2 nutrients-15-00852-f002:**
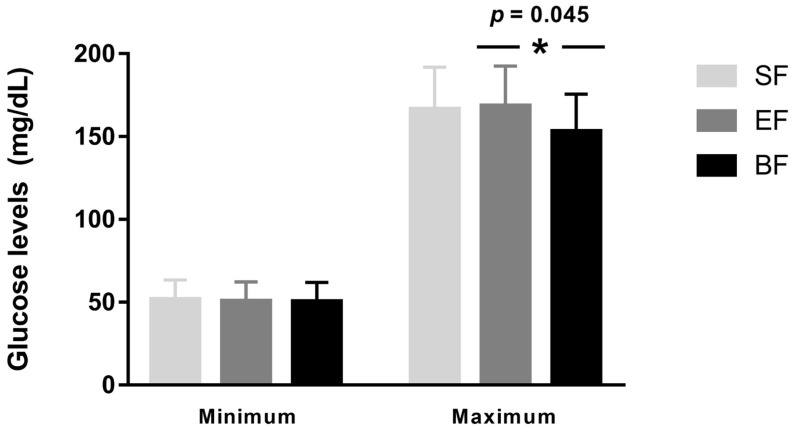
Minimum and maximum means of glucose levels by study group. Data are presented as mean ± SD. * *p*-value for differences between EF and BF groups. BF: breastfeeding; EF: experimental infant formula; SF: standard infant formula.

**Figure 3 nutrients-15-00852-f003:**
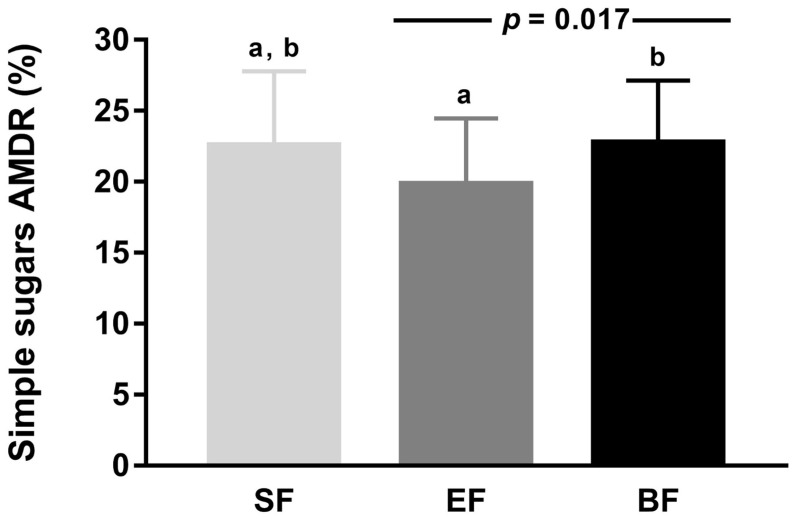
Simple sugars acceptable macronutrient distribution ranges (AMDR) in children by study groups at 6 years old. Data are presented as mean ± SD. Values which do not share the same suffix (ab) are significantly different in a Bonferroni post hoc test. *p*-value for differences between EF and BF groups. AMDR: acceptable macronutrient distribution ranges; BF: breastfeeding; EF: experimental infant formula; SF: standard infant formula.

**Figure 4 nutrients-15-00852-f004:**
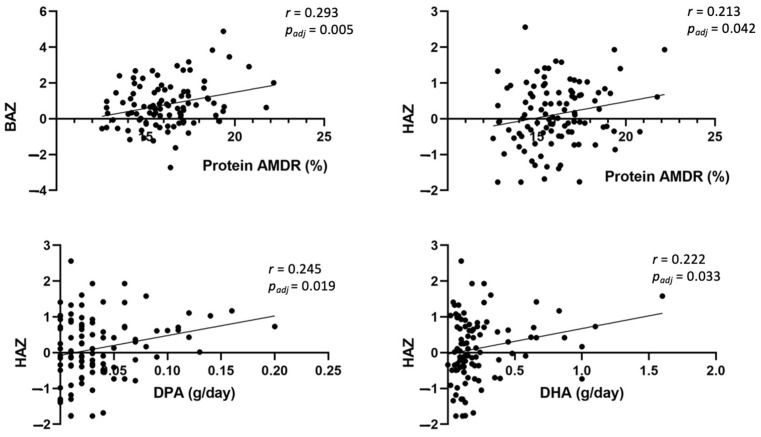
Partial correlation to evaluate potential associations between anthropometric data and dietary intake in children at 6 years, adjusted by study group, maternal age, parents’ educational level, and socioeconomic status. AMDR: acceptable macronutrient distribution ranges; BAZ: body mass index for age *z*-score; DHA: docohexaenoic acid; DPA: docosapentaenoic acid; HAZ: height for age *z*-score.

**Figure 5 nutrients-15-00852-f005:**
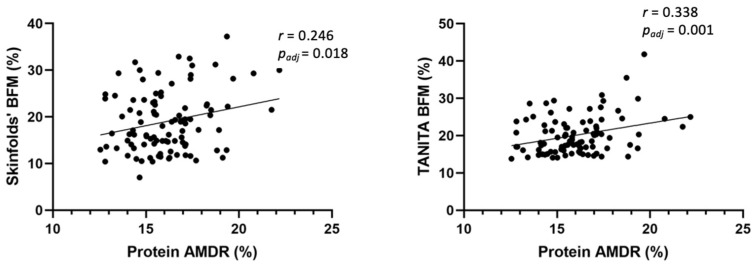
Partial correlation to evaluate potential associations between BFM and dietary intake in children at 6 years, adjusted by study group, maternal age, parents’ educational level, and socioeconomic status. AMDR: acceptable macronutrient distribution ranges; BFM: body fat mass; TANITA^®^: bioelectrical impedance.

**Figure 6 nutrients-15-00852-f006:**
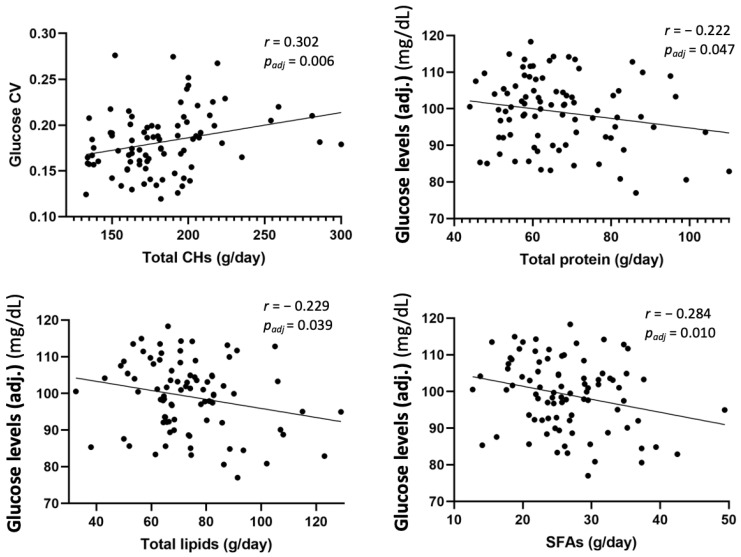
Partial correlation to evaluate potential associations between glucose data and dietary intake in children at 6 years, adjusted by study group, maternal age, parents’ educational level, and socioeconomic status. Glucose CV does not have any measurement units. Adj: adjusted mean glucose levels; CHs: carbohydrates; CV: glucose coefficient of variation; SFAs: saturated fatty acids.

**Table 1 nutrients-15-00852-t001:** Parents’ and neonatal baseline characteristics in study groups ^1^.

		SF (*n* = 32)	EF (*n* = 32)	BF (*n* = 28)	*p* ^2^
**Mother**					
Age (years)		30.61 ± 6.60 ^a,b^	30.81 ± 4.64 ^a^	34.04 ± 4.54 ^b^	**0.015**
IQ (points)		106.03 ± 12.29	102.94 ± 14.86	107.54 ± 14.63	0.43
pBMI (kg/m^2^)		24.33 (4.79)	25.28 (7.24)	24.29 (3.76)	0.74
GWG (kg)		5.60 ± 5.43	6.36 ± 5.02	6.32 ± 3.28	0.79
GDM	No	32 (100.00%)	30 (93.80%)	28 (100.00%)	0.33
	Yes	0 (0%)	2 (6.30%)	0 (0%)	
Smoking during pregnancy	No	25 (78.10%)	29 (90.60%)	26 (92.90%)	0.23
	Yes	7 (21.90%)	3 (9.40%)	2 (7.10%)	
Type of delivery	Vaginal	24 (75.00%)	22 (68.80%)	22 (78.60%)	0.68
	Cesarean	8 (25.00%)	10 (31.30%)	6 (21.40%)	
Postpartum Depression	No	24 (75.00%)	27 (87.10%)	24 (85.70%)	0.39
	Yes	8 (25.00%)	4 (12.90%)	4 (14.30%)	
Educational level	NS/Primary	2 (6.30%)	9 (29.00%)	2 (7.10%)	**0.002**
	Secondary	10 (31.30%) ^a^	7 (22.60%) ^a,b^	1 (3.60%) ^b^	
	VT	13 (40.60%)	8 (25.80%)	9 (32.10%)	
	University/ PhD	7 (21.90%) ^a^	7 (22.60%) ^a^	16 (57.10%) ^b^	
Employment status	Unemployed	6 (18.80%)	3 (9.40%)	4 (14.30%)	0.32
	Domestic work	1 (3.10%)	3 (9.40%)	0 (0%)	
	TC	2 (6.30%)	7 (21.90%)	4 (14.30%)	
	SE	23 (71.90%)	19 (59.40%)	20 (71.40%)	
**Father**					
Age (years)		32.30 ± 7.24	33.48 ± 6.43	36.36 ± 4.38	0.055
IQ (points)		108.93 ± 11.39	103.00 ± 16.75	109.28 ± 10.73	0.16
Educational level	NS/Primary	6 (18.80%) ^a,b^	13 (41.90%) ^b^	3 (10.70%) ^a^	**0.005**
	Secondary	16 (50.00%)	8 (25.80%)	7 (25.00%)	
	VT	4 (12.50%)	7 (22.60%)	6 (21.40%)	
	University/ PhD	6 (18.80%) ^a,b^	3 (9.70%) ^b^	12 (42.90%) ^a^	
Employment status	Unemployed	5 (15.60%)	2 (6.90%)	1 (3.60%)	0.53
	Domestic work	0 (0%)	0 (0%)	0 (0%)	
	TC	4 (12.50%)	2 (6.90%)	3 (10.70%)	
	SE	23 (71.90%)	25 (86.20%)	24 (85.70%)	
**Parents**					
Socioeconomic status	Low	5 (15.60%) ^a,b^	6 (18.80%) ^b^	0 (0%) ^a^	**0.004**
	Middle-Low	16 (50.00%)	14 (43.80%)	7 (25.00%)	
	Middle-High	9 (28.10%)	10 (31.30%)	11 (39.30%)	
	High	2 (6.30%) ^a^	2 (6.30%) ^a^	10 (35.70%) ^b^	
Place of residence	Urban	11 (34.40%)	11 (34.40%)	6 (21.40%)	0.46
	Rural	21 (65.60%)	21 (65.60%)	22 (78.60%)	
**Neonate**					
GA at delivery (weeks)		40.00 (2.00)	40.00 (3.00)	39.50 (3.00)	0.71
Birth Weight (kg)		3.34 ± 0.44	3.46 ± 0.53	3.39 ± 0.41	0.57
Birth Length (cm)		51.00 (3.00)	51.00 (3.60)	51.00 (2.30)	0.91
Birth HC (cm)		35.00 (2.00)	34.50 (1.10)	35.00 (2.00)	0.45
Breastfeeding (days)		14.50 (21.50) ^a^	13.50 (33.00) ^a^	390.00 (270.00) ^b^	**<0.001**
Sex	Boy	21 (65.60%)	19 (59.40%)	11 (39.30%)	0.11
	Girl	11 (34.40%)	13 (40.60%)	17 (60.70%)	
Siblings	0	7 (21.90%)	7 (22.60%)	4 (14.30%)	0.68
	≥1	25 (78.10%)	24 (77.40%)	24 (85.70%)	

^1^ Parametrically distributed data are presented as mean ± SD, categorical data as *n* (%), and non-parametrically distributed data as median (IQR). ^2^
*p*-values for overall differences between study groups. ANOVA was carried out for normally distributed variables, Kruskal–Wallis test for non-normal continuous variables, and Chi-square or Fisher test for categorical variables. Values not sharing the same suffix (ab) were significantly different in the Bonferroni post hoc test. Bold: *p*-values < 0.05. BF: breastfeeding; EF: experimental infant formula; GA: gestational age; GDM: gestational diabetes mellitus; GWG: gestational weight gain; HC: head circumference; IQ: intelligence quotient; IQR: interquartile range; NS: no schooling; pBMI: pre-conceptional body mass index; SE: stable employment; SF: standard infant formula; TC: temporary contract; VT: vocational training.

**Table 2 nutrients-15-00852-t002:** Children’s anthropometric characteristics at 6 years old ^1^.

Parameter		SF (*n* = 32)	EF (*n* = 32)	BF (*n* = 28)	*p* ^2^	*p_adj_* ^3^
Age (years)		6.08 (0.11)	6.08 (0.14)	6.06 (0.11)	0.76	-
BMI (kg/m^2^)		16.19 (2.68)	16.40 (2.42)	16.02 (2.21)	0.86	0.65
Weight (kg)		22.74 ± 4.19	22.57 ± 3.59	22.95 ± 3.71	0.93	0.80
Height (cm)		116.76 ± 4.43	115.86 ± 4.65	117.42 ± 4.84	0.43	0.80
BAZ		0.61 ± 1.28	0.80 ± 1.14	0.66 ± 1.02	0.80	0.93
WAZ		0.51 ± 1.22	0.55 ± 1.05	0.63 ± 1.08	0.91	0.90
HAZ		0.09 ± 0.86	−0.04 ± 0.84	0.28 ± 0.94	0.38	0.91
HC (cm)		51.71 ± 1.75	51.63 ± 1.60	51.68 ± 1.15	0.98	0.26
TS (mm)		9.80 (4.80)	9.90 (4.70)	9.84 (4.45)	0.93	0.84
SS (mm)		5.70 (2.68)	6.05 (1.95)	5.83 (2.10)	0.94	0.85
Skinfolds’ BFM (%)		16.18 (10.35)	16.32 (7.84)	20.36 (9.66)	0.89	0.96
Skinfolds’ BFM classification	Thin	1 (3.20%)	0 (0%)	0 (0%)	0.87	-
	Normoweight	22 (71.00%)	25 (78.10%)	21 (75.00%)		
	Overweight	5 (16.10%)	3 (9.40%)	5 (17.90%)		
	Obesity	3 (9.70%)	4 (12.50%)	2 (7.10%)		
TANITA^®^ BFM (%)		19.41 ± 4.57	19.56 ± 4.59	21.04 ± 4.36	0.35	0.35
TANITA^®^ BFM classification	Thin	0 (0%)	0 (0%)	0 (0%)	0.99	-
	Normoweight	21 (65.60%)	21 (67.70%)	16 (64.00%)		
	Overweight	6 (18.80%)	5 (16.10%)	5 (20.00%)		
	Obesity	5 (15.60%)	5 (16.10%)	4 (16.00%)		

^1^ Parametrically distributed data are presented as mean ± SD, categorical data as *n* (%), and non-parametrically distributed data as median (IQR). ^2^
*p*-values for overall differences between study groups. ^3^
*p*-values for overall differences between study groups adjusted by maternal age, parents’ educational level, and socioeconomic status through a multivariate analysis. ANOVA was carried out for normally distributed variables, Kruskal–Wallis test for non-normal continuous variables, and Chi-square or Fisher test for categorical variables. Adj: adjusted; BAZ: body mass index for age *z*-score; BF: breastfeeding; BFM: body fat mass; EF: experimental infant formula; HAZ: height for age *z*-score; HC: head circumference; IQR: interquartile range; SF: standard infant formula; SS: subscapular skinfold; TANITA^®^: bioelectrical impedance; TS: triceps skinfold; WAZ: weight for age *z*-score.

**Table 3 nutrients-15-00852-t003:** Baseline glucose data at 6 years old considering study groups ^1^.

Glucose Data	SF (*n* = 32)	EF (*n* = 32)	BF (*n* = 28)	*p* ^2^	*p_adj_* ^3^
Mean glucose levels (mg/dL)	95.98 ± 9.38 ^a^	94.13 ± 9.08 ^a,b^	90.10 ± 8.26 ^b^	**0.040**	**0.026**
Adjusted mean glucose levels (mg/dL)	101.80 ± 9.50 ^a^	99.99 ± 9.05 ^a,b^	94.96 ± 8.78 ^b^	**0.015**	**0.005**
3′–30′ MSE increment	0.194 ± 0.089 ^a^	0.236 ± 0.096 ^a^	0.254 ± 0.104 ^a^	**0.048**	0.081
Glucose CV	0.185 ± 0.037 ^a,b^	0.190 ± 0.035 ^a^	0.169 ± 0.025 ^b^	**0.046**	**0.014**
Glucose CV (%)	18.49 ± 3.69 ^a,b^	19.02 ± 3.55 ^a^	16.92 ± 2.46 ^b^	**0.046**	**0.014**

^1^ Parametrically distributed data are presented as mean ± SD. ^2^
*p*-values for overall differences between study groups. ^3^
*p*-values for overall differences between study groups adjusted by maternal age, parents’ educational level, and socioeconomic status through a multivariate analysis. ANOVA was carried out for normally distributed variables. Values not sharing the same suffix (ab) were significantly different in the Bonferroni post hoc test. Bold: *p*-values < 0.05. A 3′–30′ MSE increment does not have any measurement units. Adj: adjusted; BF: breastfeeding; CV: glucose coefficient of variation; EF: experimental infant formula; 3′–30′ MSE increment: increment of multiscale sample entropy at 3–30 min; SF: standard infant formula.

**Table 4 nutrients-15-00852-t004:** Children’s MSE at 6 years old according to their type of feeding during the first 18 months of age.

Minutes	SF (*n* = 32)	EF (*n* = 32)	BF (*n* = 28)	*p* ^1^	*p* ^2^	*p* ^3^
MSE 3′	0.175 ± 0.057	0.186 ± 0.047	0.196 ± 0.043	0.40	0.11	0.40
MSE 6′	0.217 ± 0.071	0.233 ± 0.059	0.249 ± 0.059	0.31	0.059	0.30
MSE 9′	0.244 ± 0.084	0.267 ± 0.074	0.285 ± 0.072	0.25	**0.045**	0.33
MSE 12′	0.270 ± 0.094	0.296 ± 0.086	0.321 ± 0.089	0.25	**0.034**	0.28
MSE 15′	0.290 ± 0.106	0.319 ± 0.097	0.343 ± 0.100	0.24	**0.048**	0.35
MSE 18′	0.304 ± 0.109	0.344 ± 0.105	0.364 ± 0.107	0.14	**0.037**	0.47
MSE 21′	0.316 ± 0.109	0.363 ± 0.117	0.388 ± 0.113	0.10	**0.016**	0.41
MSE 24′	0.342 ± 0.124	0.390 ± 0.126	0.407 ± 0.120	0.13	**0.045**	0.59
MSE 27′	0.361 ± 0.128	0.407 ± 0.129	0.428 ± 0.134	0.15	0.054	0.56
MSE 30′	0.369 ± 0.132	0.423 ± 0.135	0.450 ± 0.132	0.11	**0.021**	0.45

Data are presented as mean ± SD. *p*-values for overall differences between study groups: *p*
^1^ SF vs. EF; *p*
^2^ SF vs. BF; *p*
^3^ EF vs. BF. Bold: *p*-values < 0.05. MSE does not have any measurement units. BF: breastfeeding; EF: experimental infant formula; MSE: multiscale sample entropy; SF: standard infant formula.

**Table 5 nutrients-15-00852-t005:** MSE analysis in children considering growth velocity according to weight gain from birth to 6 months of age.

Minutes	NWGV (*n* = 57)	RWGV (*n* = 12)	SWGV (*n* = 19)	*p* ^1^	*p* ^2^	*p* ^3^
MSE 3′	0.191 ± 0.050	0.159 ± 0.041	0.195 ± 0.040	**0.026**	0.75	**0.023**
MSE 6′	0.238 ± 0.066	0.205 ± 0.047	0.243 ± 0.053	0.058	0.71	**0.047**
MSE 9′	0.271 ± 0.082	0.232 ± 0.052	0.277 ± 0.066	**0.045**	0.75	**0.045**
MSE 12′	0.301 ± 0.096	0.256 ± 0.056	0.312 ± 0.076	**0.037**	0.63	**0.026**
MSE 15′	0.326 ± 0.108	0.270 ± 0.063	0.331 ± 0.083	**0.025**	0.81	**0.029**
MSE 18′	0.345 ± 0.113	0.288 ± 0.069	0.357 ± 0.095	**0.030**	0.65	**0.025**
MSE 21′	0.363 ± 0.119	0.298 ± 0.071	0.380 ± 0.109	**0.019**	0.58	**0.017**
MSE 24′	0.388 ± 0.131	0.321± 0.074	0.405 ± 0.110	**0.022**	0.57	**0.017**
MSE 27′	0.410 ± 0.139	0.347 ± 0.082	0.409 ± 0.112	**0.045**	0.98	0.085
MSE 30′	0.424 ± 0.142	0.364 ± 0.094	0.432 ± 0.116	0.085	0.81	0.087
Adj. glucose	97.83 ± 9.51	102.13 ± 7.45	101.06 ± 10.33	0.10	0.24	0.74

Data are presented as mean ± SD. *p*-values for overall differences between weight gain velocity groups: *p*
^1^ normal vs. rapid; *p*
^2^ normal vs. slow; *p*
^3^ Rapid vs. Slow. Bold: *p*-values < 0.05. MSE does not have any measurement units. Adj. glucose: adjusted mean glucose levels; MSE: multiscale sample entropy; NWGV: normal weight gain velocity; RWGV: rapid weight gain velocity; SWGV: slow weight gain velocity.

**Table 6 nutrients-15-00852-t006:** MSE analysis by study group in six-year-old normoweight (NW) children, calculated using the Slaughter’s equations.

Minutes	SF (*n* = 22)	EF (*n* = 25)	BF (*n* = 21)	*p* ^1^
MSE 3′	0.167 ± 0.056 ^a^	0.176 ± 0.043 ^a^	0.197 ± 0.047 ^a^	**0.045**
MSE 6′	0.206 ± 0.070 ^a^	0.221 ± 0.054 ^a^	0.251 ± 0.065 ^a^	**0.019**
MSE 9′	0.233 ± 0.081 ^a^	0.253 ± 0.068 ^a^	0.288 ± 0.078 ^a^	**0.018**
MSE 12′	0.257 ± 0.090 ^a^	0.281 ± 0.077 ^a,b^	0.325 ± 0.096 ^b^	**0.012**
MSE 15′	0.274 ± 0.100 ^a^	0.302 ± 0.088 ^a,b^	0.352 ± 0.105 ^b^	**0.011**
MSE 18′	0.287 ± 0.101 ^a^	0.325 ± 0.094 ^a,b^	0.373 ± 0.108 ^b^	**0.006**
MSE 21′	0.299 ± 0.103 ^a^	0.340 ± 0.102 ^a,b^	0.398 ± 0.114 ^b^	**0.003**
MSE 24′	0.323 ± 0.113 ^a^	0.364± 0.103 ^a,b^	0.419 ± 0.123 ^b^	**0.007**
MSE 27′	0.337 ± 0.113 ^a^	0.388 ± 0.116 ^a,b^	0.441 ± 0.131 ^b^	**0.006**
MSE 30′	0.346 ± 0.116 ^a^	0.403 ± 0.121 ^a,b^	0.464 ± 0.134 ^b^	**0.002**

Data are presented as mean ± SD. ^1^
*p*-values for overall differences between study groups: SF, EF, and BF. ANOVA was carried out. Values which do not share the same suffix (ab) are significantly different in a Bonferroni post hoc test. Bold: *p*-values < 0.05. MSE does not have any measurement units. BF: breastfeeding; EF: experimental infant formula; MSE: multiscale sample entropy; SF: standard infant formula.

## Data Availability

The raw data supporting the conclusions of this article will be made available by the authors, without undue reservation.

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
