# Peer review of "Long-Term Effects and Potential Impact of Early Nutrition with Breast Milk or Infant Formula on Glucose Homeostasis Control in Healthy Children at 6 Years Old: A Follow-Up from the COGNIS Study"

_nutrients, 2023, doi:10.3390/nu15040852_

Round 1

Reviewer 1 Report

Dear Authors, 

I have read with interest your paper on long-term effects of early nutrition on glucose homesostasis control in healthy children at 6 years old.

The topic is of paramount importance given the need of information about the impact of early nutrition on later develpment of metabolic diseases in order to design nutritional prevention strategies. 

The study is a complex one and you made a really deep analysis of a great amount of data. 

I have just a copule of simple remarks. 

In the "material and methods "section, it is not so clear for me when the three-days dietray records have been collected. Concomintantly with CGM device or in a another moment?

Results are well presented even if not always so simple to be quickly understood, probably due to their overall complexity. You can maybe try to simplify or shorten the results section. Tables and graphs are complete and useful. 

Discussion section is well written and informative, maybe just a little too long. Strenghts and limitations of the study are clearly presented. 

Author Response

Dear Reviewer, thank you very much for your careful and thoughtful revision and comments for our manuscript. We really appreciate them and consider that they will contribute to improve the quality of the article.

We use the “Track Changes” function in Microsoft Word for the revisions. The review option used in the manuscript was: “Show Revisions in Balloons”.

Reviewer 2 Report

Nutrients-2185483

Long-term effects of early nutrition on glucose homeostasis control in healthy children at 6 years old: A follow-up from the COGNIS study

General comments

The authors provide a very interesting and important manuscript on glucose homeostasis among children who were initially feed two different infant formulas or were fed human milk.  In general, this is a well-written manuscript.

Since this early nutrition via infant formula and human milk is the thrust of the research, then the manuscript title should reflect this feeding regimen along with the intent to assess the potential impact the early feeds on glucose homeostasis at 6 years. 

Abstract

L31 – this like should state lower “fasting” glucose or was this “x” hrs pre-meal part of CGM?

L33 – the glib remark on functional similarities of EF and BF is not supported in this section of the manuscript.

L36 – growth velocity of BF infants is not new

Introduction

L47 – do not use contractions in a scientific manuscript, such as etc on this line

L51 – should include a comment that over nutrition during this early lactational period contributes to adipose hyperplasia; here’s a recent reference to this important point: Arner P. Fat Tissue Growth and Development in Humans. Nestle Nutr Inst Workshop Ser. 2018;89:37-45. doi: 10.1159/000486491. Epub 2018 Jul 10. PMID: 29991030.

L94 – while the author cite Bender et al (2014), the word “prevent” is not acceptable; the phrase should be “could help reduce the risk of metabolic diseases”

L97 – similarly here, should read “a target for disease risk reduction”.

Materials and Methods

L120 – is MFGM really a bioactive nutrient?  Not a nutrient, but perhaps a bioactive component (see work by many investigators, such as Brink LR, Lönnerdal B. Milk fat globule membrane: the role of its various components in infant health and development. J Nutr Biochem. 2020 Nov;85:108465. doi: 10.1016/j.jnutbio.2020.108465. Epub 2020 Aug 3. PMID: 32758540.; Cavaletto M, Givonetti A, Cattaneo C. The Immunological Role of Milk Fat Globule Membrane. Nutrients. 2022 Oct 31;14(21):4574. doi: 10.3390/nu14214574. PMID: 36364836; PMCID: PMC9655658; Gázquez A, Sabater-Molina M, Domínguez-López I, Sánchez-Campillo M, Torrento N, Tibau J, Moreno-Muñoz JA, Rodríguez-Palmero M, López-Sabater MC, Larqué E. Milk fat globule membrane plus milk fat increase docosahexaenoic acid availability in infant formulas. Eur J Nutr. 2022 Oct 25. doi: 10.1007/s00394-022-03024-5. Epub ahead of print. PMID: 36280613.

L155 – excellent approach to assess anthropometrics of study participants, and follow-up calculations of z-scores

L173 – excellent description on the assessment of BFM

Statistics

L232 – the indicated stat tools are acceptable; good explanations that provide clarity to the authors’ stat approaches

Results

This is a well-written section.

L278 – as the authors discuss the anthropometrics of study participants, do they have any anthropometric data on their parents?

L339 – the reported observation on infant growth are similar to those observed by this reviewer many times decades ago.

Discussion

L459 – kudos to the authors; section is well-written; authors demonstrated excellent critical thinking on the presented data and findings.

L495 – good discussion on potential mechanisms of action (MOA) while addressing carbohydrates and metabolic programming à L504 good words of caution pertinent to the hypothesis

L571 – as the authors discuss protein, they are reminded that the highest protein requirement per kg bw is during infancy; then there is the general public that is enamored with dietary protein, especial plant-based proteins…which have their own metabolic adequacy challenges when feeding infants and children.  Plant-based protein is considered low quality…poor digestibility and low EAA

L578 – the comments on glycemic index are interesting; the USA has not adopted any GI assessment; the USA dietary guidelines has not addressed this topic, which is really quite complex (see paper by Pi-Sunyer FX. Glycemic index and disease. Am J Clin Nutr. 2002 Jul;76(1):290S-8S. doi: 10.1093/ajcn/76/1.290S. PMID: 12081854.

L630 – good points on study limitations

Conclusion

L636 – good narrative

L637 – remove “prevention”; dietary and lifestyle interventions à reduce risks of developing NCD (non-communicable disease)

L654 – good comments on future infant formulas; many clinical issues and regulatory statutes to address; there is not any international harmonization with these two points

Tables

Table 1 – very clear; well done

Table 2 – excellent

Table 3 – good; don’t need the extra significant figures on clinical data

Tables 4, 5, 6 – very interesting; good presentation of data…

Figures

Figure 1 – Excellent

Figure 2 – are these glucose min/max levels in the continuum of CGM?

Figure 3 – the key statement pertinent to this figure is on L413; this is likely to be missed by the casual reader and media outlets

Figures 4, 5,6 – some of these appear to present statistical outliers which should be discussed in the narrative

Author Response

We really appreciate the comments arisen from the Reviewer and we are thankful for them as they would improve for sure our manuscript. A point-to-point answers follow your questions and comments is attached.

We use the “Track Changes” function in Microsoft Word for the revision. The review option used in the manuscript was: “Show Revisions in Balloons”.
